# Sports Analysis of Wheelchair Basketball Game Statistics

**Víctor Hernández-Beltrán** [1,*], **Sergio J. Ibáñez** [1], **Mário C. Espada** [2,3,4,5,6] **and José M. Gamonales** [1,7,8,*]

1 Training Optimization and Sports Performance Research Group (GOERD), Faculty of Sport Science, University of Extremadura, 10005 Cáceres, Spain; sibanez@unex.es
2 Instituto Politécnico de Setúbal, Escola Superior de Educação, 2914-504 Setúbal, Portugal; mario.espada@ese.ips.pt
3 Life Quality Research Centre (CIEQV-Leiria), Complexo Andaluz, 2040-413 Santarém, Portugal
4 Centre for the Study of Human Performance (CIPER), Faculdade de Motricidade Humana, Universidade de Lisboa, 1499-002 Lisbon, Portugal
5 Comprehensive Health Research Centre (CHRC), Universidade de Évora, 7004-516 Évora, Portugal
6 SPRINT Sport Physical Activity and Health Research & Innovation Center, Centro de Investigação e Inovação em Desporto Atividade Física e Saúde, 2001-904 Santarém, Portugal
7 Programa de Doctorado en Educación y Tecnología, Universidad a Distancia de Madrid, 28400 Madrid, Spain
8 Faculty of Education and Psychology, University of Extremadura, 06006 Badajoz, Spain
\* Correspondence: vhernandpw@alumnos.unex.es (V.H.-B.); martingamonales@unex.es (J.M.G.)

**Featured Application: Analyzing the performance of Wheelchair Basketball players according to Functional Classification and game indicators to provide relevant information regarding the identification of details which can determine indicial and team success in each of the different Functional Classifications.**

**Abstract:** Game statistics are used for the analysis of the performance of athletes. This allows the strengths and weaknesses of the players to be understood, and to design and implement training sessions for performance enhancement. Therefore, the objective of this study was to analyse the game statistics of the 2021 and 2023 European Champions and the 2023 World Champions in Wheelchair Basketball (WB), both male and female. For this purpose, the game statistics of all the championships were collected, and the differences according to gender and competition analysed through a statistical and inferential analysis, considering the functional classification of the players, based on the Mann–Whitney U test for the comparison of gender, and the Kruskal–Wallis H test for the comparison between championships. The results showed that players with a classification higher than 3.0 have higher three-point and two-point throws, while players with a classification of 1.0 have lower effectiveness in shooting. Depending on the championship, a greater number of three-point throws and respective effectiveness was found in the 2023 World Championships, with the male athletes presenting a greater number of points. This information will support WB coaches in designing and implementing training sessions and patterns of play and movements in competitive moments to improve individual and team performance.

**Keywords:** paralympic; functional classification; world championship; European championship; performance

## 1. Introduction

The analysis of game statistics is a growing line of research in sports. It allows researchers, coaching teams and athletes to understand the differences in performance between teams depending on whether they play at home or away in a competition, or details related to the final result of the match [1]. At the same time, the individual and collective (team) analysis of the players should be considered as a performance indicator, making it possible to improve the training processes [2]. The analysis of the actions of the game and the statistics derived from them are included in the notational analysis [3].

Furthermore, it is conceived as a fundamental tool to improve the training process, as well as the feedback given to players to improve their sporting performance [4]. This analysis has been developed in different sports disciplines such as handball to identify the differences between the winning and losing teams, with the winning teams showing higher values in throws, blocks and counterattacks [5]. Similarly, differences in competitions regarding statistics have been analysed as a function of playing surface in tennis players [6], and in rugby, the winning teams are those with the highest number of touches, playing in the opposition half, and scoring attempts [7].

In the same way, this notational analysis of game statistics has been carried out in conventional basketball. Conventional basketball is characterised by the presence of three playing positions (point guard, forward and centre), with point guards having the highest values in offensive situations, power forwards in defensive situations and small forwards having the highest number of intermediate values [8]. Similarly, analysing patterns of play through statistics allows the most relevant technical–tactical aspects to be identified and to adapt the game to these situations in order to improve competitive performance [9]; one of these examples was the evidence that winning teams have fewer ball possessions and better offensive and defensive ratios [10]. Therefore, to improve game control and position specificity, the coaching staff must identify player profiles [11]. However, the coaching staff must consider other aspects which influence sports performance like teamwork and motivation [12], leadership skills [13] or the influence of feedback [14,15]. In order to improve the team's performance, the leader of the group must develop a higher cognitive ability to mediate with their teammates and enhance the team's performance [16]. For this reason, the coaching staff must consider all these variables in order to improve the team's performance during matches.

In Wheelchair Basketball (WB), analysis has been performed to recognize the characteristics and game profiles of the players according to their Functional Classification (FC) [17]. A player's FC depends on several factors, such as the range of motion of the trunk in the three planes (frontal, lateral and vertical), as well as the injury suffered (spinal cord injury, amputees, spina bifida, joint or muscle limitations), limiting their functional and movement capacity [18]. Therefore, players receive a score between 1.0, for those players with the lowest FC, and 4.5, for those players with the highest range of motion [19], and a total of 14.5 points in the quintet on the court cannot be exceeded. Specifically in WB game statistics, Gómez et al. [20] identified that shooting percentage and free throws were the most important factors in men's games. On the contrary, in women's games, offensive rebounding and field goal percentage were the most important variables.

Moreover, Vanlandewijck et al. [21] identified that 1.0 CF players made fewer assists, with 3.0 FC players performing a greater number of blind blocks in attacks. Furthermore, another study found that 4.0 FC players made the highest number of three-point throws, with a 42% success rate; in addition, they registered the highest maximum values in offensive and defensive rebounds, receiving an average of 9.95 fouls per game, and scoring 29.4 points [22]. Therefore, the greater the similarity and closeness in the FC of the players, the more similar the sports performance will be, increasing the throw efficiency [23]. In the same way, a significant correlation has been identified between the number of successful throws, assists and steals as a function of the starting five in women's WB [24]. Through the study of throws, the coaching staff can obtain relevant information about the player's performance in terms of efficiency, and, consequently, identify the areas with the highest percentage of success, as well as which FC has the highest efficiency [25]. Shooting in basketball is one of the main indicators of the sporting performance of WB players, both in training and in official competitions. Therefore, the coaching staff should take into consideration this technical skill and moments of the game to define the ideal quintet to play on the court.

There is a great disparity in the results and specificity in each of the specific positions or FC of each player. Therefore, it is of vital importance to be able to optimize and analyse performance using specific training focused on the playing techniques associated with the

demand of the player's position. Therefore, the present study aimed to analyse the game statistics of different international WB competitions (European Championships 2021, World Championships 2022 and European Championships 2023). The secondary objectives were (i) to identify significant differences in the performance of WB players according to the FC considering the game statistics, (ii) to analyse the game profiles according to the gender of the athletes and (iii) to evaluate the performance of the players according to the different competitions analysed.

## 2. Materials and Methods

### 2.1. Design

The design of this study was empirical, using a descriptive and associative strategy [26]. It can also be classified as ex post facto research, where data analysis was carried out retrospectively [27]. In addition, as no manipulation of any of the study variables was carried out, it is considered a naturalistic study [28].

### 2.2. Sample

The study sample consisted of a total of 5238 cases, divided into 3255 cases corresponding to men's matches and 1983 to women's matches. Likewise, a total of 1452 cases were identified in the 2021 European Championship, 2247 cases in the 2023 World Championship and 1539 cases in the 2023 European Championship.

Finally, considering the players' FC, the sample was divided into FC1 ($n = 961$), FC1.5 ($n = 445$), FC2 ($n = 510$), FC2.5 ($n = 562$), FC3 ($n = 755$), FC3.5 ($n = 469$), FC4 ($n = 880$) and FC4.5 ($n = 645$).

### 2.3. Procedures

Firstly, a process was carried out to compile the game statistics for the three competitions. These data were then entered into a Microsoft Excel spreadsheet (v. 2006, Microsoft Corporation for Mac, Redmond, WA, USA) to carry out descriptive and inferential analyses according to the dependent variables established.

### 2.4. Variables

The variables collected for the analysis of throws in the WB competitions were as follows:

- Dependent variables:
  - ○ FC of the players.
  - ○ Competition analysed.
  - ○ Player's gender.

- Independent variables:
  - ○ Points.
  - ○ Attempted free throws.
  - ○ Successful free throws.
  - ○ Attempted 2-point field goals.
  - ○ Successful 2-point field goals.
  - ○ Attempted 3-point field goals.
  - ○ Successful 3-point field goals.

### 2.5. Statistical Analysis

The Kolmogorov–Smirnov test [29] was used to analyse the sample distribution, obtaining a value lower than $p < 0.05$. Therefore, nonparametric tests were used for data analysis. A descriptive analysis (mean and standard deviation) was carried out to characterise the sample considering the dependent variable.

Secondly, the differences between the dependent variables were analysed considering the player's FC and the competition. For this purpose, the nonparametric Kruskal–Wallis H

test was used. Moreover, through pairwise comparisons, significant differences between the dependent variables and the Eta square ($\eta_p^2$) were calculated for each analysis to determine the effect size and were interpreted as $\eta_p^2 < 0.01$ trivial, $\eta_p^2 = 0.01$ to $0.06$ low, $\eta_p^2 = 0.06$ to $0.14$ moderate, and $\eta_p^2 > 0.14$ high [30]. Likewise, to calculate the differences between the players' gender, the Mann–Whitney U test was used.

Statistical analysis was performed using the Statistical Package for the Social Sciences software (v. 27, 2021; IBM Corp., IBM SPSS Statistics for MAC OS, Armonk, NY, USA) and significance was accepted at $p \leq 0.05$.

## 3. Results

Table 1 shows the results related to the descriptive and inferential analysis considering the competition. The 2023 European Championship is the one that presents the highest mean values in most of the variables studied, and, therefore, it is the competition that presents the greatest differences from the rest of the competitions.

**Table 1.** Descriptive and inferential analysis considering the competition.

| Variables | 2021 European Championship | | 2023 European Championship | | 2023 World Championship | | H | Df | *p* | Post Hoc | Eta Square |
|---|---|---|---|---|---|---|---|---|---|---|---|
| | $\overline{X}$ | SD | $\overline{X}$ | SD | $\overline{X}$ | SD | | | | | |
| Points | 4.48 | 6.10 | 5.09 | 6.60 | 4.92 | 6.82 | 7.21 | 2 | 0.027 * | a | 0.027 * |
| Total successful throws | 1.97 | 2.79 | 2.24 | 2.89 | 2.14 | 3.01 | 8.15 | 2 | 0.017 * | a | 0.018 * |
| Total attempted throws | 4.72 | 5.30 | 5.11 | 5.50 | 4.96 | 5.95 | 7.14 | 2 | 0.028 * | b | 0.048 * |
| Successful 2-point field goals | 1.86 | 2.66 | 2.13 | 2.78 | 2.03 | 2.89 | 9.20 | 2 | 0.010 * | a | 0.010 * |
| Attempted 2-point field goals | 4.24 | 4.73 | 4.72 | 5.04 | 4.43 | 5.36 | 10.89 | 2 | 0.004 * | a b | 0.023 * 0.006 * |
| Successful 3-point field goals | 0.12 | 0.47 | 0.11 | 0.41 | 0.11 | 0.44 | 0.20 | 2 | 0.901 | N/A | |
| Attempted 3-point field goals | 0.48 | 1.24 | 0.39 | 1.01 | 0.53 | 1.42 | 0.89 | 2 | 0.641 | N/A | |
| Successful free throws | 0.48 | 1.48 | 0.50 | 1.15 | 0.51 | 1.16 | 1.87 | 2 | 0.392 | N/A | |
| Attempted free throws | 0.89 | 1.70 | 0.93 | 1.81 | 0.92 | 1.80 | 0.59 | 2 | 0.743 | N/A | |

H: Kruskal–Wallis H test; Df: Degree of freedom; * $p \leq 0.05$; a: European Championship 2021–European Championship 2023; b: European Championship 2023–World Championship 2023; N/A: Not applicable.

Table 2 shows the descriptive and inferential results of the throws made by the players according to gender. Considering the significant differences, differences are observed in most of the variables except for total and two-point throws attempted, since they take a very similar number of throws. Therefore, if we consider the rest of the variables analysed, statistically significant differences are observed according to gender ($p < 0.05$).

**Table 2.** Descriptive and inferential analysis considering the players' gender.

| Variables | Male | | Female | | Mann–Whitney U Test | Standardised Test Statistic | *p* |
|---|---|---|---|---|---|---|---|
| | $\overline{X}$ | SD | $\overline{X}$ | SD | | | |
| Points | 5.07 | 6.57 | 4.47 | 6.54 | 3,025,075.000 | −3.957 | 0.000 * |
| Total successful throws | 2.19 | 2.85 | 2.02 | 3.01 | 3,064,696.500 | −3.204 | 0.001 * |
| Total attempted throws | 4.93 | 5.48 | 4.95 | 5.91 | 3,176,085.000 | −0.979 | 0.328 |
| Successful 2-point field goals | 2.04 | 2.69 | 1.97 | 2.96 | 3,111,749.500 | −2.281 | 0.023 * |
| Attempted 2-point field goals | 4.32 | 4.82 | 4.70 | 5.52 | 3,268,993.500 | 0.796 | 0.426 |
| Successful 3-point field goals | 0.15 | 0.52 | 0.04 | 0.25 | 3,011,420.500 | −8.776 | 0.000 * |
| Attempted 3-point field goals | 0.61 | 1.43 | 0.25 | 0.89 | 2,809,618.500 | −11.191 | 0.000 * |
| Successful free throws | 0.54 | 1.17 | 0.43 | 1.38 | 3,057,416.500 | −4.326 | 0.000 * |
| Attempted free throws | 0.96 | 1.80 | 0.85 | 1.74 | 3,101,465.500 | −2.919 | 0.004 * |

* $p \leq 0.05$.

Table 3 presents the descriptive and inferential analysis of the players' FC, along with the variables that exhibit statistical differences between the FC. It was observed that all the variables show significant differences, but players with higher FC tend to have better performance in terms of throwing and points.

**Table 3.** Descriptive and inferential analysis regarding the players' FC.

| Variables | FC1 | | FC1.5 | | FC2 | | FC2.5 | | FC3 | | FC3.5 | | FC4 | | FC4.5 | | H | Df | $p$ |
|---|---|---|---|---|---|---|---|---|---|---|---|---|---|---|---|---|---|---|---|
| | $\overline{X}$ | SD | $\overline{X}$ | SD | $\overline{X}$ | SD | $\overline{X}$ | SD | $\overline{X}$ | SD | $\overline{X}$ | SD | $\overline{X}$ | SD | $\overline{X}$ | SD | | | |
| Points | 1.39 | 2.43 | 2.53 | 4.20 | 3.88 | 5.22 | 4.08 | 5.10 | 5.66 | 6.28 | 5.05 | 6.49 | 7.57 | 8.22 | 8.20 | 8.40 | 641.056 | 7 | 0.000 * |
| Successful throws | 0.65 | 1.15 | 1.17 | 1.96 | 1.75 | 2.38 | 1.80 | 2.22 | 2.47 | 2.76 | 2.15 | 2.85 | 3.27 | 3.61 | 3.57 | 3.85 | 592.291 | 7 | 0.000 * |
| Attempted throws | 1.66 | 2.41 | 2.77 | 3.51 | 4.38 | 4.88 | 4.65 | 4.46 | 5.72 | 5.785 | 5.16 | 5.75 | 7.31 | 6.68 | 7.67 | 6.75 | 662.148 | 7 | 0.000 * |
| Successful 2-point field goals | 0.64 | 1.14 | 1.16 | 1.94 | 1.65 | 2.25 | 1.68 | 2.04 | 2.28 | 2.60 | 2.03 | 2.76 | 3.10 | 3.50 | 3.41 | 3.73 | 562.713 | 7 | 0.000 * |
| Attempted 2-point field goals | 1.63 | 2.34 | 2.65 | 3.35 | 3.92 | 4.44 | 4.12 | 3.78 | 5.02 | 5.10 | 4.60 | 5.16 | 6.57 | 6.10 | 7.03 | 6.25 | 619.868 | 7 | 0.000 * |
| Successful 3-point field goals | 0.01 | 0.09 | 0.02 | 0.14 | 0.10 | 0.40 | 0.12 | 0.48 | 0.18 | 0.55 | 0.12 | 0.46 | 0.17 | 0.54 | 0.16 | 0.53 | 149.511 | 7 | 0.000 * |
| Attempted 3-point field goals | 0.04 | 0.30 | 0.11 | 0.41 | 0.45 | 1.09 | 0.53 | 1.27 | 0.70 | 1.51 | 0.56 | 1.60 | 0.74 | 1.60 | 0.64 | 1.33 | 349.281 | 7 | 0.000 * |
| Successful free throws | 0.08 | 0.33 | 0.16 | 0.57 | 0.26 | 0.72 | 0.35 | 0.92 | 0.54 | 1.11 | 0.62 | 1.21 | 0.85 | 1.44 | 1.04 | 2.23 | 490.064 | 7 | 0.000 * |
| Attempted free throws | 0.21 | 0.70 | 0.36 | 0.99 | 0.51 | 1.20 | 0.70 | 1.39 | 1.01 | 1.71 | 1.07 | 1.73 | 1.51 | 2.17 | 1.82 | 2.63 | 538.983 | 7 | 0.000 * |

H: Kruskal–Wallis H test; Df: Degree of freedom; * $p \leq 0.05$.

## 4. Discussion

Generally, the players' game-related statistics varied according to playing position, probably because of the well-known differences in the players' FC, which conditionate the distance they play from the basket. The present study obtained significant results considering the competition, gender and FC of the players. In the analysed competitions, differences were identified in the number of points, as well as in the number of two-point throws attempted and respective successful rate, with higher values found in the 2023 European Championship. Taking the gender of the athletes as a reference, male players showed higher results and considering the players' FC, significant differences were observed in all the analysed variables, with players associated with higher functionality developing higher values. This fact reinforces previous studies, with FC4 players shooting more and obtaining higher scores [22].

Through the study of throws, the coaching staff can obtain relevant information about the player's performance in terms of efficiency, and, consequently, understand the areas associated with the highest percentage of success, as well as identify which FC has the highest efficiency. Coaches can use these results to reinforce the importance of relying on different players' contributions to team performance and evaluate players' game performance according to their playing position. Thus, these discriminant models could help in player recruitment and improving training programmes [2].

The results according to the competition analyses reported that there are significant differences depending on the analysed competition, with the 2023 European Championship being the one with the best results in terms of the throws and points, with each player scoring an average of five points per game. Taking two-point throws into consideration, there are differences between the analysed European Championships, with an average of 2.13 converted throws and 4.72 attempted throws in the 2023 European Championships. Similarly, comparing the 2023 European Championships and the World Championships, the results show that the World Championships obtained lower values, with an average of 2.03 throws converted and 4.43 throws attempted. These results show that the 2023 European Championships were one of the most closely contested competitions, with the highest percentage of both throws and scores in each of the matches. These results could be produced because a change in the game's pattern, the competition's inherent quality or the players' abilities. On the other hand, no differences were identified in the number of three-point throws, which is in contrast to previous results in the literature, since three-point throws are closely related to winning the match [31]. By analysing tournament statistics,

coaches can identify trends, evaluate team performance and plan successful strategies to win the game.

As for the results obtained when taking the gender of the players as a reference, male athletes ($n$ = 5.07) scored a greater number of average points per game than female athletes ($n$ = 4.47). In the same way, considering two- and three-point throws, as well as free throws, the male players exhibited the best results. On the other hand, after analysing the results in an individual perspective, the total number of throws and two-point throws attempted was not associated with significant differences between the genders. These results are in agreement with previous studies carried out in basketball, where male athletes showed greater efficiency and number of throws taken during competitions [32]. These differences in performance are related to anthropometric, functional and motor differences between male and female teams [33]. Therefore, game dynamics and technical–tactical aspects are different depending on gender. In the same way, aspects such as the percentage of blocks and turnovers, as well as the number of missed throws, are determinants in the differences in performance between male and female players [34]. Therefore, to increase efficiency in shooting, offensive game dynamics focused on allowing a free throw to the shooter, as well as the development of organized and stable attacking systems, should be considered.

Finally, and focusing on the FC of the players, it was observed that players with a higher functionality obtained better results compared to those with a lower FC. These results are linked to the influence of FC on movement ability, as well as on agility and sprinting ability [35], since the players with a higher FC present higher values in upper body strength, directly influencing the ability to shoot at the baskets [36]. Therefore, the greater the similarity of the FCs of the players on the court, the greater the performance of the team [23]. The results obtained in the present study agree with those obtained by Pérez-Tejero and Arbex [22], with FC4 players having the highest percentage of throws and hits compared to the rest of the players. FC has been extensively studied in the field of WB since it is one of the variables with the greatest influence on sports performance, influencing the playing position [21], the final classification of the team in the competition [23] and the playing time developed by each athlete [37]. Considering these results, the coaching staff must develop tasks to improve the shooting efficiency of the players and optimize their sport performance. Players must increase their efficiency in three-point shots and free throws because these are the shots with the largest influence on the final score.

In the same way, the definition of the starting five of the matches should be carefully considered, because it directly influences the final result of the match [24]. Therefore, it can be concluded that the greatest differences in performance occur between those players with greater differences in FC, with players with similar FC being those who present better performance and physical condition [17]. Hence, the coaching staff should consider the demands of the competitions, as well as the playing positions of each athlete, to understand and evaluate the shooting possibilities of each player and strengthen this aspect, aiming not only for individual performance enhancement but also team performance improvement in WB.

Some of the limitations that arose during the development of the study were related to the scarcity of scientific studies that analyse statistics in WB. This fact made it difficult to compare the results obtained with previous research focused on the object of study. In the same way, the lack of equality in the sample regarding gender or the FC is a limitation of this study, because some categories could be over-represented or under-represented. In turn, the origin of these results must be analysed, as well as how they influence the game, since it may be due to an evolution in playing patterns, or to the limitation and reduction of the technical–tactical skills of the players. As for the strengths, the results are very powerful, because they allow us to determine the efficiency of the players' throws according to their FC, gender or competitions. This provides great information to the coaching staff to develop sessions focused on those positions that present a lower throwing efficiency. For future studies, it is recommended to carry out an observational analysis of throwing in a high-level competition, to determine those areas with greater efficiency, as

well as finding out the most decisive playing positions in the final result using throwing as a reference.

## 5. Conclusions

Analysing the performance of WB players according to the FC and game indicators provide relevant information regarding the identification of details which can determine indicial and team success in each of the different FCs. This analysis enables coaching staff and athletes to improve daily strategies in training and establish a gameplay pattern based on the competition itself. Based on these results, the coaching staff must devise tasks that will enhance the players' shooting efficiency and improve their overall performance. The players should focus on improving their efficiency in three-point shots and free throws since these have a greater impact on the final score.

The coaching staff should comprehend which players have a higher probability of success in shooting to create game systems and playing opportunities to isolate these players and facilitate their shooting possibilities. Therefore, in this case, the FC3 and FC4 players should be considered because they perform the greatest number of two- and three-point throws and are the most effective. In contrast, the FC1 players can oversee making transitions and facilitating the movements of teammates by making blocks. They are the players with the lowest efficiency in shooting due to their lower functionality and limited movement in their trunks.

**Author Contributions:** Conceptualization: V.H.-B., S.J.I. and J.M.G.; methodology: V.H.-B., S.J.I. and J.M.G.; formal analysis: V.H.-B., S.J.I. and J.M.G.; investigation: V.H.-B., S.J.I. and J.M.G.; supervision: V.H.-B., S.J.I. and J.M.G.; data curation: V.H.-B., S.J.I., M.C.E. and J.M.G.; writing—original draft preparation: V.H.-B., S.J.I., M.C.E. and J.M.G.; writing—review and editing: V.H.-B., S.J.I., M.C.E. and J.M.G.; visualisation: V.H.-B., S.J.I., M.C.E. and J.M.G.; funding acquisition: V.H.-B., S.J.I., M.C.E. and J.M.G. All authors have read and agreed to the published version of the manuscript.

**Funding:** This study was supported by the Portuguese Foundation for Science and Technology, I.P. under Grant UID04045/2020 and Instituto Politécnico de Setúbal. Also, the research was partially funded by the GOERD of University of Extremadura and the Research Vice-rectory of Universidad Nacional. This study has been partially supported by the funding for research groups (GR21149) granted by the Government of Extremadura (Employment and infrastructure office—Consejería de Empleo e Infraestructuras), with the contribution of the European Union through the European Regional Development Fund (ERDF) by the Optimisation of Training and Sports Performance Research Group (GOERD) of the Faculty of Sports Sciences of the University of Extremadura.

**Institutional Review Board Statement:** Not applicable.

**Informed Consent Statement:** Not applicable.

**Data Availability Statement:** The data supporting this study's findings are available from the corresponding author upon reasonable request.

**Acknowledgments:** The authors would like to acknowledge the participants who allowed us to conduct this study.

**Conflicts of Interest:** The authors declare no conflict of interest.

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
