# Peer review of "Sports Analysis of Wheelchair Basketball Game Statistics"

_applsci, doi:10.3390/app14072923_

Round 1

Reviewer 1 Report

Comments and Suggestions for Authors

The manuscript is well-written and addresses adequately the observed topic. However, I have some general comments and suggestions for improving the quality of the article.

Introduction

Include more specific examples or case studies to illustrate the impact of analyzing game statistics on improving performance. For instance, including a detailed case study where a basketball team improved their performance by analyzing their game statistics and making targeted changes to their training regimen can help illustrate the effectiveness of this analytical approach.

Integrate opposing viewpoints or potential limitations of using game statistics analysis to provide a more balanced perspective. Including a section that discusses the limitations of relying solely on game statistics for performance evaluation, such as overlooking intangible qualities like teamwork or leadership skills, can provide a more nuanced view of the topic.

Materials and Methods

This section is well developed and detailedly explained.

Discussion

Discuss the implications of the study findings on coaching strategies and player development more explicitly.  By elaborating on how coaches can use the study results to tailor training programs to improve player efficiency in shooting and optimize team performance, the article can highlight the actionable insights derived from the analysis.

Integrate a section addressing potential limitations of the study in more detail. Providing a more thorough discussion on the limitations, such as the scarcity of existing research in the field of basketball statistics analysis, can offer transparency and context to the study's constraints. Expanding on the limitations section to discuss how the scarcity of scientific studies in basketball statistics analysis impacted the ability to compare and validate the study results can provide a clearer understanding of the study's challenges.

Author Response

Thank you so much for your reports. All the comments have been considered deeply and modified. A letter with the answer to each review comment has been added. 

Reviewer 2 Report

Comments and Suggestions for Authors

Sports Analysis of Wheelchair Basketball Game Statistics

Rewiew

The research on "Sports Analysis of Wheelchair Basketball Game Statistics" is relevant and up-to-date for several reasons:

Recent Data Analysis: The study focuses on game statistics from the 2021 and 2023 European Champions and the 2023 World Champions in Wheelchair Basketball. Analyzing the most recent competitions ensures that the findings reflect current trends, tactics, and player performance levels, which are crucial for developing current training and game strategies.

Inclusive and Diverse Perspective: By including both male and female athletes in wheelchair basketball, the research addresses the importance of gender diversity in sports science. This inclusivity ensures that the training sessions and competitive strategies developed from this research are applicable and beneficial to a wider range of athletes.

Performance Optimization: The research aims to identify the strengths and weaknesses of players based on game statistics. This targeted analysis helps in designing specific training sessions that can address identified weaknesses and enhance players' strengths, leading to performance optimization.

Innovative Statistical Analysis: Utilizing statistical and inferential analysis methods, such as the Mann Whitney U test for gender comparisons and the Kruskal Wallis H test for comparing different championships, provides a robust framework for understanding the impact of various factors on player performance. This methodological approach can serve as a model for future sports analytics research.

Strategic Development for Coaches: The findings offer practical implications for wheelchair basketball coaches by highlighting effective patterns of play and movements. This strategic insight is vital for developing training programs that can adapt to the evolving dynamics of wheelchair basketball and enhance both individual and team performance.

Focus on Functional Classification: By considering the functional classification of the players, the research acknowledges the importance of understanding how different levels of physical ability impact game performance. This approach is particularly relevant in para-sports, where classification plays a crucial role in ensuring fair and competitive play.

Contribution to Wheelchair Basketball and Para-sports: Finally, the research contributes valuable knowledge to the field of wheelchair basketball and para-sports in general. By advancing our understanding of game statistics and their implications for performance, the study supports the growth and professionalization of wheelchair basketball.

In summary, this research is relevant and up-to-date due to its focus on recent data, inclusive approach, methodological rigor, and its practical implications for improving performance in wheelchair basketball. It addresses a need for current, data-driven insights that can directly impact training and competitive strategies in para-sports.

The methodology of the research on the analysis of game statistics in wheelchair basketball incorporates several commendable practices, including the use of empirical, descriptive, and associative strategies, as well as naturalistic observation without manipulation of variables. However, like any study, this methodology is not without potential criticisms. Key areas where the methods could face criticism include:

  1. Sample Distribution and Selection: The division of cases between men's and women's matches, as well as among different functional classification (FC) categories, might not be representative of the wider population of wheelchair basketball athletes. Critics could argue that the sample distribution might lead to biases in the findings, especially if certain categories are overrepresented or underrepresented in the sample.
  2. Dependence on Self-Reported Data: If any part of the data collection relied on self-reported measures (not explicitly mentioned but a common issue in similar studies), it could introduce bias and inaccuracies into the dataset. This could be a point of criticism, particularly concerning the accuracy and reliability of the collected data.

This should be written in the limitation section!

Result:

The main criticism of the result presented, particularly regarding the 2023 European Championship data, appears to revolve around its significantly higher mean values in most variables compared to other competitions. This distinction suggests that the 2023 European Championship is markedly different in terms of player performance metrics such as points scored, successful throws, and attempted throws, among others.

A critical perspective might question the comparability of these competitions or suggest that the distinctiveness of the 2023 European Championship could be attributed to factors not directly related to the competition's inherent quality or the players' abilities. These could include variations in competition conditions, the caliber of teams or players participating, or even methodological differences in how data was collected or analyzed.

Furthermore, the statistical significance of these differences, indicated by p-values less than 0.05 across multiple categories, underscores the need for a careful examination of what these differences imply about the competition itself and the factors contributing to these outcomes. The criticism could extend to discussing whether these differences highlight a trend in the evolution of the sport or if they reflect inconsistencies in the competitive landscape that could affect the generalization of these results to future competitions or broader sport analytics.

Criticism of the results must be reflected in the article or limitation!

I recommend to publish after minor corrections.

Author Response

(The authors gave the same response as above.)
